# The Functional and Application Possibilities of Starch/Chitosan Polymer Composites Modified by Graphene Oxide

**DOI:** 10.3390/ijms23115956

**Published:** 2022-05-25

**Authors:** Magdalena Krystyjan, Gohar Khachatryan, Karen Khachatryan, Anna Konieczna-Molenda, Anna Grzesiakowska, Marta Kuchta-Gładysz, Agnieszka Kawecka, Wiktoria Grzebieniarz, Nikola Nowak

**Affiliations:** 1Faculty of Food Technology, University of Agriculture in Krakow, Al. Mickiewicza 21, 31-120 Krakow, Poland; gohar.khachatryan@urk.edu.pl (G.K.); karen.khachatryan@urk.edu.pl (K.K.); anna.konieczna-molenda@urk.edu.pl (A.K.-M.); wiktoria.grzebieniarz@urk.edu.pl (W.G.); nikola.nowak@urk.edu.pl (N.N.); 2Faculty of Animal Science, University of Agriculture in Krakow, Al. Mickiewicza 21, 31-120 Krakow, Poland; anna.grzesiakowska@urk.edu.pl (A.G.); marta.kuchta-gladysz@urk.edu.pl (M.K.-G.); 3Department of Product Packaging, Cracow University of Economics, ul. Rakowicka 27, 30-149 Krakow, Poland; kaweckaa@uek.krakow.pl

**Keywords:** starch, chitosan, graphene oxide, nanoparticles, polysaccharides, composites, polymers

## Abstract

This study describes functional properties of bionanocomposites consisting of starch/chitosan/graphene oxide (GO) obtained using the green synthesis method, such as water-barrier and optical properties, as well as the rate of degradation by enzymatic and acid hydrolysis. The toxicity of the composites and their effects on the development of pathogenic microflora during storage of meat food products was also investigated. Although the results showed that the barrier properties of the composites were weak, they were similar to those of biological systems. The studies carried out confirmed the good optical properties of the composites containing chitosan, which makes it possible to use them as active elements of packaging. The susceptibility of starch and chitosan films to enzymatic and acid hydrolyses indicates their relatively high biodegradability. The lack of toxicity and the high barrier against many microorganisms offer great potential for applications in the food industry.

## 1. Introduction

The day we succeeded in obtaining plastic became a watershed moment that revolutionized the industry. Today, there is probably no area of our lives where this material is not used. It has become an indispensable part of our lives and accompanies us at every step. However, the intensive development and opportunities associated with the use of synthetic materials have negatively affected the environment due to the limited biodegradability of these materials [1]. The world is facing problems with the disposal of waste materials, which are polluting nature and also pose a threat to our health. Both global and local action is needed to counteract these adverse effects. Therefore, one of the objectives of the European Green Deal that the European Commission has put forward in 2019 is sustainable waste management [2].

In the last decade, there has been a rapidly growing interest in materials from renewable sources that are fully biodegradable and do not harm the environment. Among these, plant material plays an important role as it can be used to produce raw materials with a wide range of application possibilities. Starch, cellulose, alginates and chitosan are among the most widely used raw materials due to their availability, ease of acquisition, low production costs and unique properties, which have led to their use in many industries [3,4]. The films made from these materials, however, have certain limitations—i.e., low barrier properties and poor resistance to mechanical damage—and because they are biological materials they are also subject to rapid deterioration, which inhibits their wider application. A solution to this problem may be the use of composite systems, both hydrophilic and hydrophobic, which would improve the barrier properties of such composites. The addition of antimicrobial substances would prevent spoilage of biodegradable packaging while also protecting the packaged product. Numerous studies have confirmed that this direction is absolutely right [4,5,6,7,8,9,10].

Scientific research on biological materials of “nano” dimensions is a very current topic. Functional nanomaterials have been especially appreciated in medicine, as they offer excellent prospects for the development of new, non-invasive methods of diagnosing and treating cancer. Nanomaterials can be used for monitoring the progress of therapy or disease, determining blood type in patients requiring transfusions or for tissue typing in transplantation [11,12,13]. Among the natural polymers used to create nanocomposites, polysaccharides have attracted particular attention. They are available, biodegradable and safe for humans; they have film, foaming and gel-forming properties; and they provide a barrier against oxygen and carbon dioxide [3,5,14,15,16,17]. Depending on the origin, polymers may exhibit many other desirable attributes, such as antibacterial, antioxidant, antioxidant, immunomodulatory, anti-inflammatory and antiviral properties. As a consequence, materials produced on this basis have a very wide range of applications [8,18,19,20,21].

Graphene is one of the many allotropic varieties of carbon, consisting of a single layer of graphite in which the carbon molecules form a hexagonal arrangement [21]. It is so far the strongest material discovered by man, stronger than steel of the same thickness and more resistant than diamond. It has high thermal and electrical conductivity, provides a barrier against gases and has the ability to absorb visible and near-infrared light [22,23]. It can also be easily stretched and coated on the surface of various materials and in different forms, and recent studies have also confirmed its bactericidal and bacteriostatic properties [6,24,25]. These unique properties mean that graphene has great potential and it has already been successfully applied in many fields, such as the automotive industry, aerospace, electrical engineering, robotics, solar cells, energy storage, telecommunications, biochemistry and medicine [22,26,27,28,29,30,31,32,33,34]. These advantages make it possible to use graphene for stabilization and structure reinforcement of polymers that have certain limitations; i.e., natural polymers such as starch, chitosan, cellulose, alginates, hyaluronic acid, which, despite their many advantages, show weak barrier properties or poor mechanical properties. On the other hand, natural polymers, thanks to their attributes such as low production costs, biodegradability, renewability, lack of toxicity and high compatibility with various materials, can be good carriers of nanoparticles [5,35,36,37,38,39,40,41].

With the above in mind, we proposed a method for the green synthesis of novel starch/chitosan/GO bionanocomposites that improved numerous physicochemical and biological properties, making them a promising alternative to purely synthetic materials. The results of this study were published in our earlier article [6]. The developed composites exhibited interesting properties. Cell-based analyses revealed no toxic effect from the nanocomposites on HaCat keratinocytes and HepG2 hepatoma cells, although a pronounced bacteriostatic effect against various strains of pathogenic bacteria was observed. Therefore, the aim of this study was to extend the scope of the research to fully characterize the innovative nature of nanocomposites and determine the applicability of such materials. The scope of the study included investigating the barrier properties of the composites and to determining to what extent they would protect food from spoilage. The study also aimed to check the degradation rate of the nanocomposites in enzymatic and acid hydrolysis reactions, investigate their optical properties and show possible sensitivity to environmental changes. In addition, the toxicity levels of the nanocomposites and their antimicrobial activity were improved.

## 2. Results and Discussion

### 2.1. Water Vapour Transmission Rate

The results for the water vapour transmission rate are collected in Table 1. The conducted tests showed that the WVTR for the tested composites was high, which entails limited usefulness due to a low water vapour barrier. Plastic films are characterized by a barrier several times higher [42]. However, the value obtained for the index is in line with the values obtained among biodegradable polymers, the WVTR value of which can reach 2900 (g/m^2^·d) [43]. Since thin composites showed better barrier properties, they were used for further investigation.

### 2.2. Enzymatic and Acid Hydrolysis of Composites

The susceptibility of starch–chitosan films to enzymatic hydrolysis was studied in two reactions: enzymatic hydrolysis of chitosan using chitosanase and enzymatic hydrolysis of starch using glucoamylase. Figure 1 shows the dependence of the concentration of reducing sugars per glucosamine on the reaction time of the enzymatic hydrolysis using chitosanase.

Using linear regression and the dependence of log(1/c) on time, t, the rate constants for the successive steps of the enzymatic reaction were obtained from the linear regression correlation R = 0.97–0.99 (Figure 2). The reaction appeared to be a first-order process (Table 2). The starch hydrolysis reaction was divided into steps with a linear dependence of the concentration of reducing sugars on the reaction time. Starch hydrolysis catalysed by glucoamylase proceeds in two steps (Figure 3). The linear relationship suggests a zero-order hydrolysis reaction using α-amylase. For each of the hydrolysis steps, the reaction rate constant k was determined from the slope coefficient of the straight line.

The acid hydrolysis of starch–chitosan films proceeds in a single step with a linear dependence of product growth on reaction time. This course of degradation suggests a zero order for the acid hydrolysis reaction. The rate constants of the acid hydrolysis reaction (*ka*) were determined for each film, as a factor of the slope of the straight line [44]. The values of the rate constants are presented in Table 2.

A high efficiency for the enzymatic hydrolysis of chitosan in films was obtained by Poshina and co-workers [45], and a relatively high efficiency of starch hydrolysis showed by Castillo et al. [46]. High values for the enzymatic hydrolysis reaction rate constants prove that the obtained films are highly biodegradable. Additionally, due to their chitosan–starch composition they are environmentally friendly [47].

Both film components, i.e., starch and chitosan, were hydrolysed during a single acid hydrolysis reaction. Figure 4 shows the dependence of the concentration of reducing sugars on the reaction time of the acid hydrolysis of starch–chitosan films.

For the acid hydrolysis, very mild reaction conditions were used [48,49]. Despite the use of mild acid hydrolysis reaction conditions, significant film degradation was achieved after just 10 days [50].

The enzymatic hydrolysis reaction of nanocomposite I was the fastest followed by nanocomposite II, with the slowest degradation being composite C. This applied to the enzymatic hydrolysis of both chitosan and starch. This was confirmed by the values of the reaction rate constants. It was shown that the hydrolysis reaction of chitosan is a first-order reaction. In contrast, the hydrolysis of starch proceeds in two steps, and each step is a zero-order reaction. The acid hydrolysis of folate proceeds in a one-step zero-order reaction. In the acid hydrolysis reaction, the control film (composite C) degraded fastest, while nanocomposite I and II hydrolysed slowly with a similar (within error) reaction rate constant *ka*. The susceptibility of starch–chitosan films to enzymatic and acid hydrolysis indicates their relatively high biodegradability.

### 2.3. Toxicity Profile

The toxicity of nanographene composites was assessed using two standard tests: the cell viability assessment and the comet assay.

#### 2.3.1. Cell Viability Assessment Test

In the experiment, cell viability was assessed in the different experimental groups: a suspension of pure cells immediately after collection (pure 0 h), a suspension of cells stored for 24 h, cells exposed with a control film and cells exposed with nanocomposites I and II. Based on the collected results, there was no effect from the nanocomposites on the percentage of viable cells. Detailed data are shown in Table 3.

#### 2.3.2. Comet Test

The toxicity of the nanocomposites was assessed on the basis of a comet assay (Figure 5). For this purpose, the following numbers of cells were counted for the whole test group: for pure 0 h—543 cells; for pure 24 h—326 cells; for the control composite—509 cells, for nanocomposite I—436 cells and for nanocomposite II—426 cells. Toxicity was assessed based on the estimated parameter % tail DNA content. The mean value of the comet tail DNA percentage parameter (% tail DNA) for the 0 h clean sample was 23.43 ± 8.76, while for the 24 h clean sample it was 25.81 ± 14.31 (Table 4). The mean % tail DNA for the control foil group was 16.26 ± 12.14 and for the test foils it was 6.48 ± 8.63 for nanocomposite I and 9.39 ± 10.23 for nanocomposite II (Table 4). Based on the data collected and analysed, it was concluded that the applied concentration of nanographene in the presence of the food film did not significantly affect the level of DNA integrity disruption in cells, i.e., it was not toxic to bloodstream cells.

No genotoxic effect from the nanographene-doped films on DNA integrity disorders in whole peripheral blood lymphocytes was found on the basis of the Wistar strain mouse genome. GO exhibits properties that increase the proliferation of mammalian cells, as does chitosan, included in the tested control, I and II composites [51,52]. Such properties of the tested compounds had a stabilizing effect on the survival of the tested whole peripheral blood lymphocytes during exposure to the analysed composites. The results obtained in this study allowed us to conclude that the studied nanocomposites I and II did not exhibit cytotoxic effects.

A study by Liu et al. [53] showed that GO can activate mutagenesis processes at the molecular level. In vitro GO cytotoxicity studies have shown that GO can induce changes in normal cell morphology, lead to cell membrane damage and disrupt DNA integrity in cell nuclei [54,55]. Therefore, a comet assay was used in this study to determine the toxicity of the applied GO concentrations in nanocomposites I and II. The results obtained in this study allowed us to conclude that the characterized nanocomposites did not exhibit genotoxic effects.

Chemical modifications of chromatin resulting from exogenous genotoxic agents, such as heavy metals and their derivatives, destabilize the chromatin structure. This instability is a factor generating the formation of subsequent mutations that can lead to developmental defects, reproductive disorders, neoplastic transformation or bioaccumulation in the food chain [56]. Detection of DNA damage at the single-cell level is of great importance in fields such as toxicology, pharmacy, ecotoxicology, animal and human nutrition and environmental biomonitoring [57,58]. One tool for assessing the toxicity of an agent is the comet assay. It is a sensitive test used to measure damage and repair at the DNA level in single cells. It allows damage to be assessed in different types of cells and tissues; i.e., peripheral blood, cultured cells, cancer cells, solid tumours, semen, yeast or bacteria [59]. In our study, the test material was whole peripheral blood lymphocytes from Wistar strain mice.

### 2.4. Storage Studies

The storage trials were designed to assess the microbiological quality of poultry meat stored under manufactured films at a refrigeration temperature of approximately +4 °C. Figure 6, Figure 7, Figure 8, Figure 9 and Figure 10 show results from microbiological cultures for individual groups and species of bacteria as a function of time.

GO exhibits bactericidal activity against Gram-negative and Gram-positive bacteria [60]. Our microbiological studies confirmed the above statements, as shown in Figure 6, showing changes in the total number of microorganisms (OLD) in poultry meat during storage under manufactured composites. On the first and second days of storage, no significant differences were observed between the storage conditions, but after 48 h of storage, a significant reduction in microbial growth was shown in meat stored under composite C and (nanocomposite I and II) compared to the results of samples stored accompanied by food film (FS). We can attribute the lack of significance in the differences between the control (C) and the films with added GO (nanocomposite I and II) to the presence of chitosan in the composition of the composites. Its antimicrobial action can be attributed to three mechanisms. The first is surface ionic interaction leading to leakage of cell sap. Another mechanism is the inhibition of mRNA and protein synthesis, through its penetration into cell nuclei. The antibacterial action of chitosan also includes the formation of an external barrier through metal hematization and the reduction of the absorption of nutrients, essential for the life and growth of microorganisms [61,62,63,64]. The inhibitory effect of GO addition in composites for OLD could only be observed after 72 h and 96 h of storage, after which significant differences between the control film (C) and the composites with GO addition (I and II) were observed. This result confirmed that the use of GO had a significant effect on inhibiting overall microbial counts in poultry meat and that higher amounts had a clear effect on OLD.

Examination of Enterobacteriaceae abundance (Figure 7) showed different relationships compared to the results of OLD analyses. Significant differences in the abundance of these bacteria in meat stored under food film (FS) and the control (C) compared to composites with added GO (I, II) were already noticeable after 24 h of storage. These differences persisted consistently until the end of the study (96 h). However, the validity of using more GO in the composites was not observed due to the absence of significant differences between nanocomposite samples I and II. The mechanism of the antibacterial effect of the graphene embedded in the composite may be related to its sharp edges, which cause mechanical damage to bacterial cell membranes [65]. Akhaven et al. [66] came to the same conclusion, claiming that cell membrane damage was caused by contact with sharp graphene walls, which in turn is an effective way to inhibit the growth of Gram-positive bacteria.

In the case of *E. coli* (Figure 8) and *Campylobacter* spp. (Figure 9), after 24 h storage, significant differences in their abundances were observed between meat stored under a food film (FS) and the biocomposites (C, I and II). However, the differences in the abundances of these microorganisms in meat depending on the composite used were not statistically significant. These results may indicate a lack of inhibitory effect from the film on *E. coli* and *Campylobacter* spp. or that the amounts of GO used in the composites were not sufficient to demonstrate an inhibitory effect on these microorganisms. Research shows that the sharp ends of graphene do not damage Gram-negative bacteria [66].

The analysis of the counts of coagulase-positive staphylococci (Figure 10) showed significant differences in the counts of microorganisms in meat stored under the tested films only after 96 h. However, no differences in the abundances of these microorganisms were observed in meat under nanocomposites I and II during 72 h of storage. Films produced by Kim et al. [67] also showed inhibitory effects on *E. coli.*

### 2.5. Photoluminescence Spectroscopy

Photoluminescence spectra were measured to confirm the optical properties of the produced films and possible sensitivity to environmental changes. Figure 11, Figure 12, Figure 13 and Figure 14 show the emission spectra of the composites obtained and the changes in the emissions of the films before storage testing and after four days of storage of poultry meat samples under them.

The starch–chitosan composite exhibited significant light emissions at about 450 nm due to the emission properties of chitosan [68]. Addition of GO reduced the emission intensity of the films (Figure 11). After four days of storage of the meat under the control film (composite C), the emission of this film changed slightly, with a slight decrease in intensity and a shift of the maximum towards higher wavelengths (Figure 12). Figure 13 shows the emission spectrum of nanocomposite I. In the case of this nanocomposite, the emission intensity increased after the meat was stored under it. In contrast, for nanocomposite II (Figure 14), where the GO concentration was twice as low, the changes in emission intensity were insignificant. The observed changes during storage may have been due to decomposition of GO by microorganisms. Liu and co-workers [69] showed that graphite, GO and RGO all oxidise when exposed to bacteria, with GO and RGO oxidising to a greater extent due to more original defects, which in the case of GO leads to disintegration into small pieces.

## 3. Materials and Methods

### 3.1. Materials

A control sample (composite C) was prepared from starch and chitosan gels. The nanocomposites I and II were made of starch and chitosan gels with the addition of GO. The exact procedure for obtaining nanocomposites, as well as their structure and properties, were presented in our previous work [6]. In this work, the water vapour transmission rate of thin (0.95–0.107 mm) and thick (0.205–0.219 mm) [6] composites were checked and, for further analysis, the nanocomposites with better properties were selected for testing.

### 3.2. Water Vapour Transmision Rate

The water vapour transmission rate (WVTR) was tested in accordance with the gravimetric (dish) method [70]. A glass vessel was filled with silica gel, covered with the film under investigation and sealed. Then, it was placed in a regulated microclimatic chamber at a temperature 25 °C and 75% relative humidity. After 24 h, the vessel was weighed. WVTR was assessed on the basis of weight gain. The tested specimens were prepared and then air-conditioned in a normative manner. Each test was repeated five times. WVTR was calculated according to the formula:WVTR [g/m^2^ × d] = 240 × weight of water ÷ (surface penetration × 24)

### 3.3. Enzymatic Hydrolysis

Hydrolysis of composites was performed using enzyme chitosanase from Streptomyces sp. in buffered aqueous glycerol solution, ≥15 units/mg protein (E1%) (Chitosan N-acetylglucosaminohydrolase; EC 3. 2.1.132) (C0794 Sigma-Aldrich, Poznan, Poland), and glucoamylase from Aspergillus niger, ≥260 U/mL (Amyloglucosidase EC 3.2.1.3), in aqueous solution (A7095 Sigma-Aldrich, Poznan, Poland).

Composites of 3.0 mg were placed in 50 mL of 0.1 M acetate buffer, pH 5.5. The mixture was thermostated in a 37 °C water bath for approximately 15 min, and then chitinase solution (1.0 mL) or glucoamylase solution (0.25 mL) was added. After enzyme addition, the reaction mixture was incubated with gentle stirring at 37 °C. Samples of the reaction mixture (1.0 mL) were taken after 0, 10, 20, 40, 60, 80, 100, 120, 140, 160, 180, 200, 220 and 240 min for the determination of reducing sugars. The enzymatic hydrolysis reaction was stopped by adding 1.0 mL of DNS solution. 3,5-Dinitrosalicylic acid (DNS) in alkaline sodium potassium tartrate was used as the reagent for reducing sugars, following Southgate [71]. The reaction samples were filtered through cellulose strainers with a pore diameter of 0.2 m (Waterman) and heated at 90 °C for 5 min, then cooled to 20 °C. UV-Vis spectrophotometric analysis was performed using a Shimadzu TCC-260 spectrophotometer (Shimadzu Scientific Instruments, Kyoto, Japan). Absorbance at 520 nm was recorded. The calibration curve was prepared using D-(+)-glucosamine hydrochloride (Sigma-Aldrich, Poznan, Poland) or D-(+)-glucose (Sigma-Aldrich, Poznan, Poland) as standard. All reactions were run in duplicate

### 3.4. Acid Hydrolysis

The 3.0 mg of composites were placed in 50 mL of 0.15 M hydrochloric acid. The mixture was temperature-controlled in a water bath at 40 °C. Samples of the reaction mixture (1.0 mL) were taken after 0, 0.5, 1, 1.5, 2, 2.5, 3, 3.5, 4, 5, 6, 7, 8, 9 and 10 days for determination of the concentration of reducing sugars.

The hydrolysis reaction was stopped by adding 1 mL of alkaline DNS solution. Samples were filtered through 0.2 μm pore diameter cellulose strainers (Waterman) and assayed for reducing sugars by UV-Vis spectrophotometry (Shimadzu Scientific Instruments, Kyoto, Japan) in the range from 480to 520 nm on a Shimadzu TCC-260 spectrophotometer. The calibration curve was prepared using D-(+)-glucose (Sigma-Aldrich, Poznan, Poland) as standard [72]. All reactions were run in quadruplicate.

### 3.5. Toxicity Profile

Toxicity assessment of the composites was performed on freshly collected peripheral blood from 10 wild-type Wistar (WT) mice. Viability analysis and a comet assay were used to assess toxicity. For this purpose, whole peripheral blood cells were exposed to control and composites. Two sterilized discs cut from the tested composites were placed at the bottom of a sterile Eppendorf mixer and 150 μL of whole peripheral blood and 50 μL of RPMI-1640 culture medium (Sigma Aldrich, Poznan, Poland) were pipetted. Exposure was carried out for 24 h at room temperature. The negative control consisted of blood samples not exposed to the tested composites—clean blood, used at 0 h immediately after collection stored with RPMI 1640 medium (Roswell Park Memorial Institude 1640 Medium) for 24 h at room temperature.

#### 3.5.1. Isolation of Whole Peripheral Blood Lymphocytes for Analysis

In an Eppendorf tube, 20 μL of whole peripheral blood was mixed with 1 mL of RPMI 1640 culture medium and then layered on 200 μL of Histopaque-1077 medium (Sigma-Aldrich, Poznan, Poland). The samples were centrifuged for 3 min at 500× *g*. The separated fraction containing lymphocytes and monocytes was transferred to freshly prepared 1 mL RPMI 1640 medium. The sample was centrifuged again for 3 min at 500× *g*. At the end of the isolation, the lymphocyte pellet was resuspended in 100 μL of 1% PBS (potassium buffer solution) (Sigma-Aldrich, Poznan, Poland).

#### 3.5.2. Viability Assessment

Cell viability was assessed by staining with 0.4% trypan blue solution. Ten microliters of lymphocyte isolate and ten microliters of 0.4% trypan blue (Sigma-Aldrich, Poznan, Poland) were mixed on a basal slide and incubated for 2 min at room temperature. The 10 μL were then transferred to a Bürker chamber. Live cells—unstained and dead cells stained blue—were counted under the chamber.

#### 3.5.3. Alkaline Variant Comet Test

The evaluation of changes in nuclear DNA integrity in somatic cells was performed according to the comet assay protocol of Singh et al. [73] with modification. The 10 μL of lymphocyte isolate suspended in 75 μL of LMP agarose (low melting point) (Sigma-Aldrich, Poznan, Poland) was applied to basal slides coated with NMP agarose (normal melting point) (Sigma-Aldrich, Poznan, Poland). Lysis of the slides was carried out for 1 h in alkaline buffer (2.5 M NaCl (Sigma-Aldrich, Poznan, Poland), 0.1 M EDTANa2 (ethylenediaminetetraacetic acid disodium salt dihydrate) (Sigma-Aldrich, Poznan, Poland), 10 mM TRIS (Trizma base) (Sigma-Aldrich, Poznan, Poland) and 1% Triton X-100, pH = 10 (Sigma-Aldrich, Poznan, Poland)) at +4 °C with limited light. Electrophoresis was conducted under alkaline conditions in 30 mM NaOH buffer (Sigma-Aldrich, Poznan, Poland) with 2 mM EDTANa2, pH = 12.5 (Sigma-Aldrich, Poznan, Poland), under limited light for 20 min at 0.6 V/cm. Neutralization was carried out in 0.4 M Tris (Sigma-Aldrich, Poznan, Poland). For detection, slides were stained with ethidium bromide at a concentration of 200 μg/mL. Microscopic documentation was performed using a Zeiss Imager A2 epifluorescence microscope with AxioCam MRc5 software (Carl Zeiss, Jena, Germany). Lymphocyte damage assessment was performed using CASP 1.2.3b software (CaspLab, Poland). For each animal, 50 comets were analysed. The parameter determining the toxicity profile in the comet assay was the percentage of DNA in the tail (% of DNA in the tail, TD %).

### 3.6. Storage Test

Sterile plastic containers were used for storage testing, with five replicates for each sample (K, F1, F2), and also for the food wrap (FS). Five poultry meat samples, each weighing 1 g (±0.05 g), were lined in each container. The corresponding film was placed in the cap of each container, capped and left under refrigerated conditions at 4 °C. The material was kept for 4 days, with a daily sample (1 g) taken from each container. Microbiological analyses were performed [74,75,76,77,78,79]. TBX, PCA, PRI/RPF, MCCD and VRBG media (Biomaxima, Lublin, Poland) were used for microbiological testing.

### 3.7. Photoluminescence Spectroscopy

Photoluminescence measurements of composites were performed at room temperature using an F7000 HITACHI spectrophotometer (Hitachi High-Tech Corporation, Tokyo, Japan). A 360 nm wavelength was used for the excitation. The emission spectra of the composites were measured before the storage test and after four days of storage of poultry meat under them.

## 4. Conclusions

This work presents functional properties of new bionanocomposites, enriched with GO nanoparticles and based on a binary starch/chitosan matrix, obtained using the green synthesis method. After studying the water vapour transmission rate of thin (0.95–0.107 mm) and thick (0.205–0.219 mm) composites, the thin nanocomposites, with better barrier properties were selected for further analysis. The susceptibility of starch/chitosan-based films to enzymatic and acid hydrolyses indicates their relatively high biodegradability. Cytotoxicity studies confirmed that there was no effect from the nanocomposites on the percentage of viable cells and the applied concentration of nanographene in the composites did not significantly affect the level of DNA integrity disruption in cells; i.e., it is not toxic to blood-stream cells. High barrier properties against many microorganisms entail great potential for applications in the food industry. Under the action of nanocomposites, there was an inhibition or significant slowing down of the growth of many microorganisms, especially Enterobacteriaceae, *Escherichia coli* and *Campylobacter* spp. The study confirms that not only chitosan but also GO exhibit bactericidal activity against Gram-negative and Gram-positive bacteria. Good optical properties for the composites make it possible to use them as active elements of packaging.

## Figures and Tables

**Figure 1 ijms-23-05956-f001:**
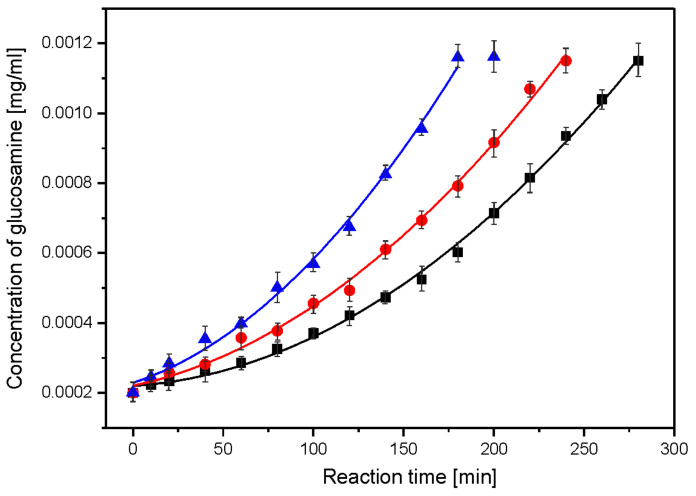
Dependence of the concentration of reducing sugars per glucosamine on the reaction time of the enzymatic hydrolysis of chitosan in composites: black—composite C, red—nanocomposite II and blue—nanocomposite I.

**Figure 2 ijms-23-05956-f002:**
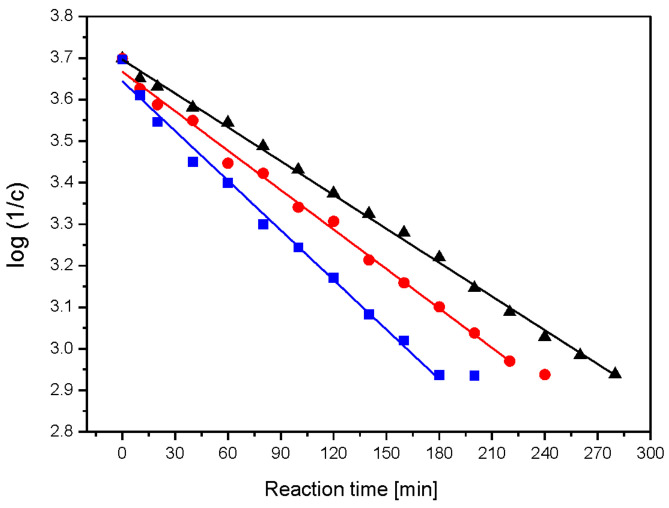
Dependence of log (1/c) on reaction time of enzymatic hydrolysis of chitosan: black—composite C, red—nanocomposite II and blue—nanocomposite I.

**Figure 3 ijms-23-05956-f003:**
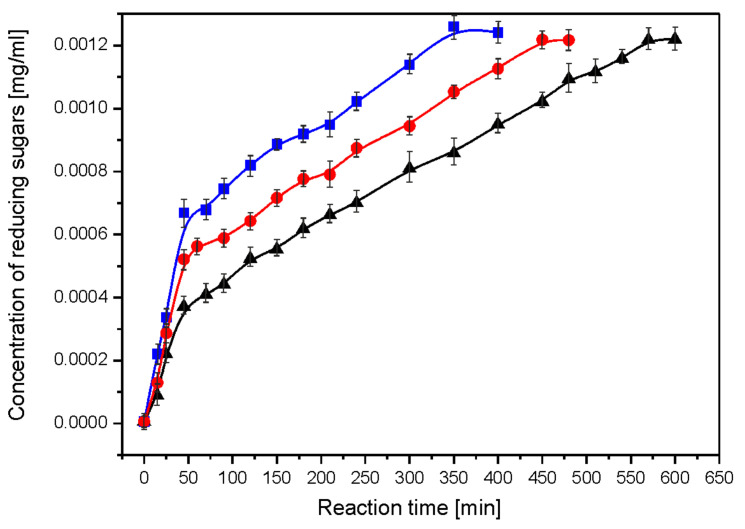
Dependence of the concentration of reducing sugars per glucose on the reaction time of the enzymatic hydrolysis of starch using glucoamylase: black—composite C, red—nanocomposite II and blue—nanocomposite I.

**Figure 4 ijms-23-05956-f004:**
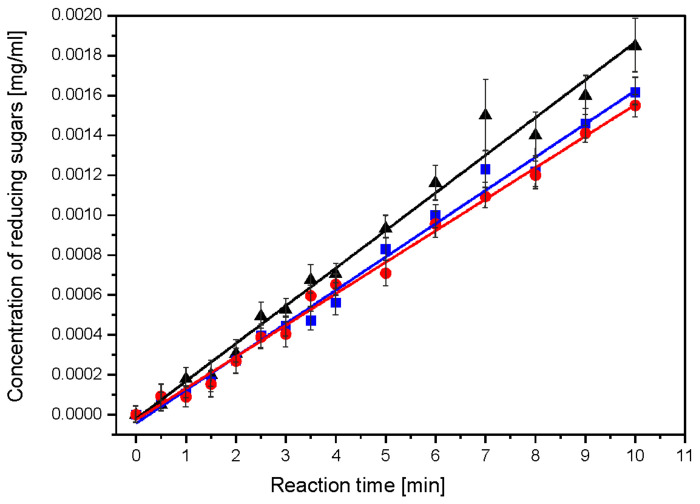
Concentration dependence of reducing sugars as a function of the acid hydrolysis reaction time for films: black—composite C, red—nanocomposite II and blue—nanocomposite I.

**Figure 5 ijms-23-05956-f005:**
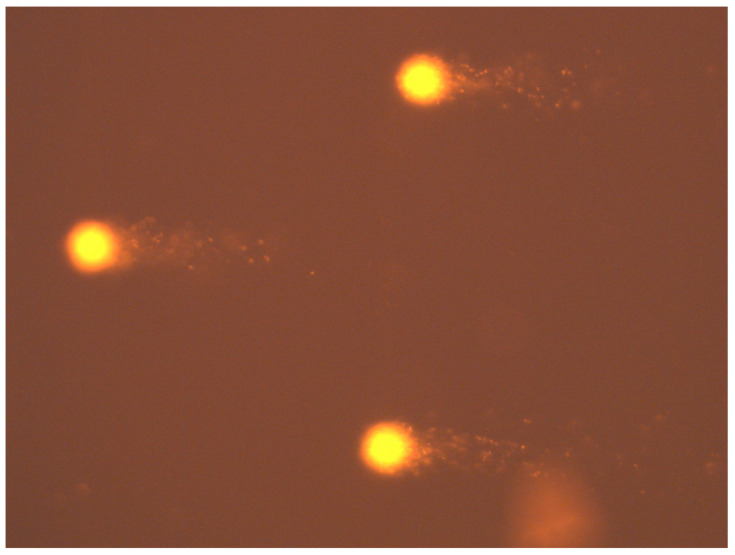
Complete peripheral blood lymphocytes after comet assay. Magnification 400×.

**Figure 6 ijms-23-05956-f006:**
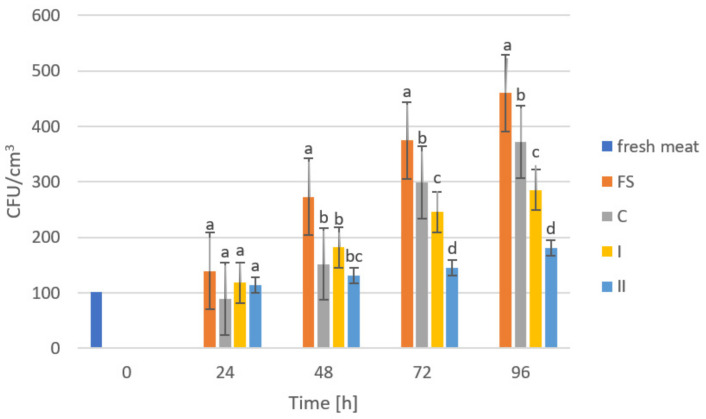
Total microbial count in poultry meat stored under different films at about 4 °C. The bars with the same letters (a, b, c, d) do not differ significantly at the level of confidence 0.05.

**Figure 7 ijms-23-05956-f007:**
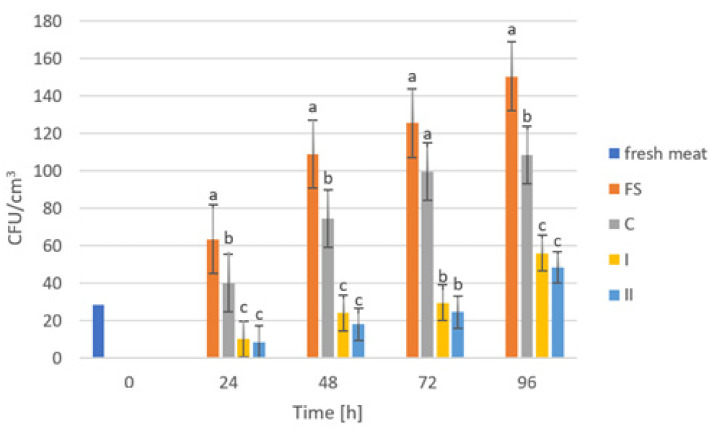
Enterobacteriaceae counts in poultry meat stored under different films at about 4 °C. The bars with the same letters (a, b, c) do not differ significantly at the level of confidence 0.05.

**Figure 8 ijms-23-05956-f008:**
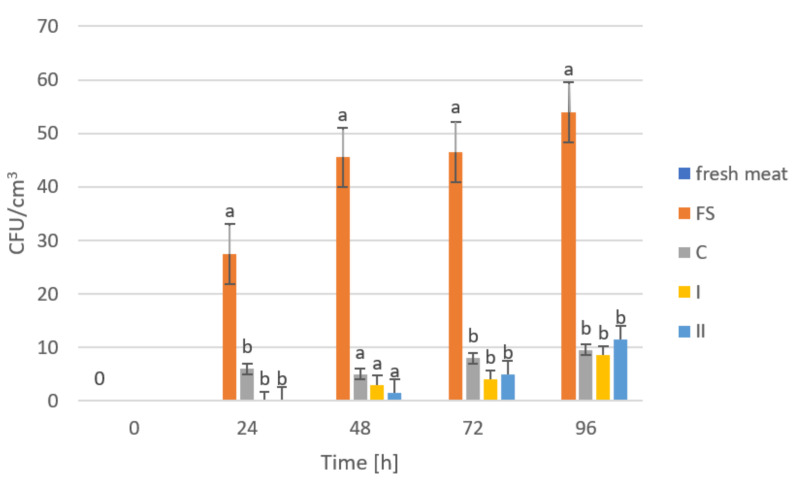
Number of *Escherichia coli* bacteria in poultry meat stored under different films at about 4 °C. The bars with the same letters (a, b) do not differ significantly at the level of confidence 0.05.

**Figure 9 ijms-23-05956-f009:**
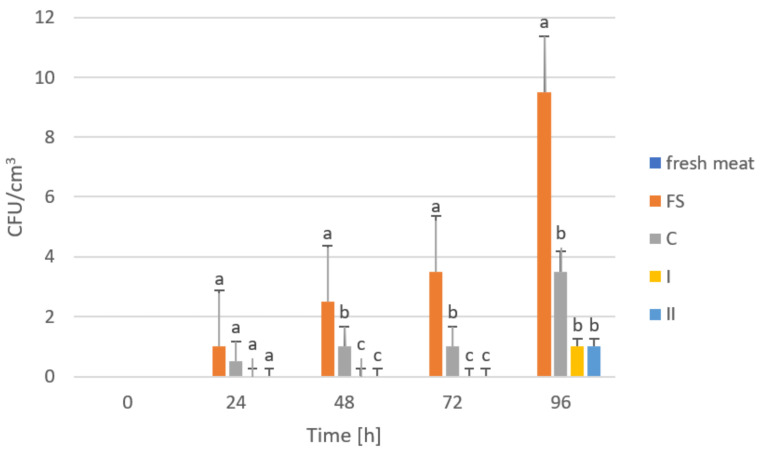
Number of *Campylobacter* spp. in poultry meat stored under different films at about 4 ℃. The bars with the same letters (a, b, c) do not differ significantly at the level of confidence 0.05.

**Figure 10 ijms-23-05956-f010:**
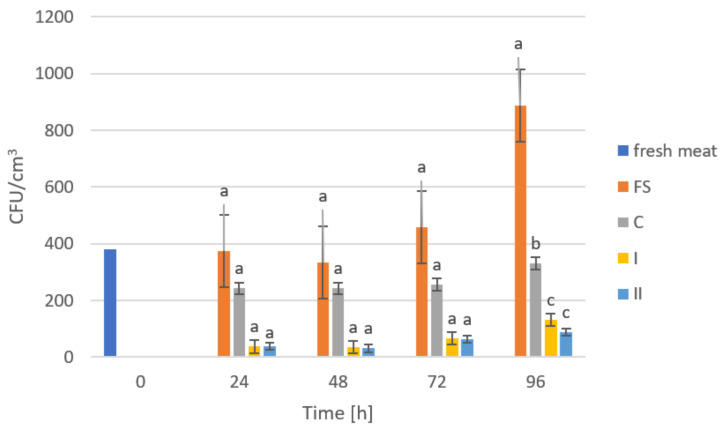
Number of coagulase-positive staphylococci in meat stored under different films at about 4 ℃. The bars with the same letters (a, b, c) do not differ significantly at the level of confidence 0.05.

**Figure 11 ijms-23-05956-f011:**
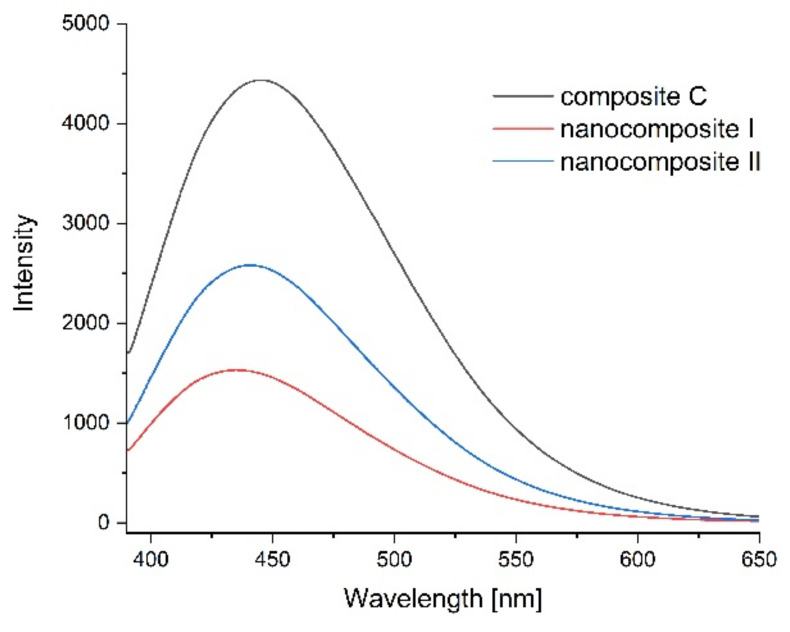
Emission spectra of control film (composite C—black line), nanocomposite I (blue line) and nanocomposite II (red line).

**Figure 12 ijms-23-05956-f012:**
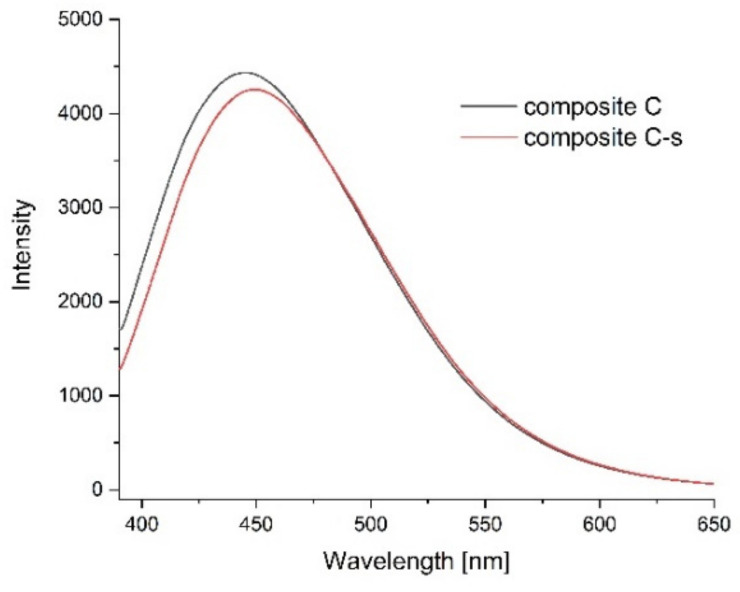
Emission spectra of control films before the storage test (composite C, black line) and after four days of storage of poultry meat under them (composite C-s, red line).

**Figure 13 ijms-23-05956-f013:**
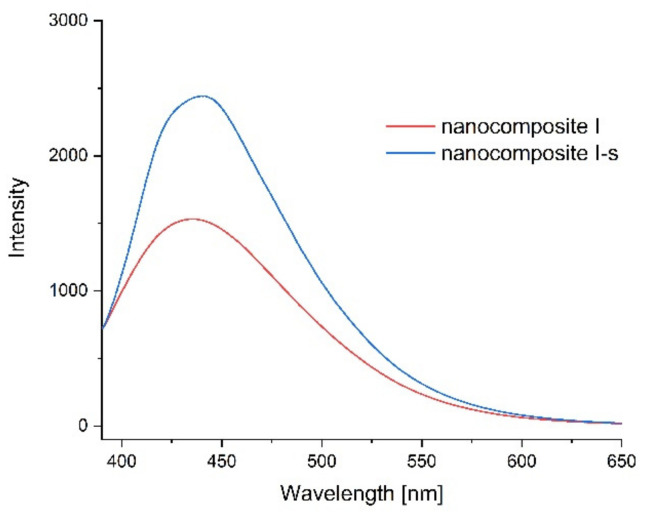
Emission spectra of nanocomposite I before the storage test (red line) and after four days of storage of poultry meat under them (composite I-s, blue line).

**Figure 14 ijms-23-05956-f014:**
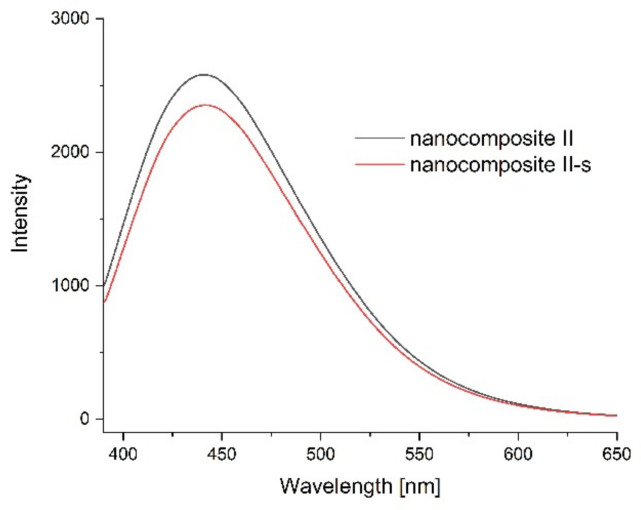
Emission spectra of nanocomposite II before the storage test (black line) and after four days of storage of poultry meat under them (composite II-s, red line).

**Table 1 ijms-23-05956-t001:** Water vapour transmission rates of composites.

Composites		WVTR (g/m^2^·d)
Composite C	Thin	742.9 ± 11.6 ^a^
Nanocomposite I	Thin	740.8 ± 8.5 ^a^
Nanocomposite II	Thin	734.7 ± 11.4 ^a^
Composite C	Thick	824.8 ±16.8 ^b^
Nanocomposite I	Thick	813.0 ± 26.3 ^b^
Nanocomposite II	Thick	787.4 ± 37.4 ^b^

The statistical Mann–Whitney U test was conducted in SPSS Statistics to check whether there were statistically significant differences in the WVTR values for the tested materials (*p* < 0.05). Parameters in columns denoted with the same letters (a, b) do not differ statistically.

**Table 2 ijms-23-05956-t002:** Values of the rate constants for the enzymatic hydrolysis and acid hydrolysis reactions.

Sample	Type of Hydrolysis			
	Enzymatic Hydrolysis of Chitosan	Enzymatic Hydrolysis of Starch		Acid Hydrolysis of Starch
	*k* × 10^−3^ (min^−1^)	*k*_1_ × 10^−6^ (mg∙mL^−1^∙min^−1^)	*k*_2_ × 10^−6^ (mg∙mL^−1^∙min^−1^)	*k_a_* × 10^−4^ (mg∙mL^−1^∙min^−1^)
Composite C	2.72 ± 0.03	8.4 ± 0.7	1.59 ± 0.02	1.88 ± 0.06
Nanocomposite I	4.00 ± 0.08	14.6 ± 0.5	1.92 ± 0.05	1.67 ± 0.04
Nanocomposite II	3.17 ± 0.06	11.7 ± 0.6	1.70 ± 0.03	1.58 ± 0.04

**Table 3 ijms-23-05956-t003:** Evaluation of cell viability.

Sample	% of Viable Cells (Average ± sd)
Non-exposed cells 0 h	95 ± 4
Non-exposed cells stored for 24 h	95 ± 4
Control composite	99 ± 2
Nanocomposite I	95 ± 6
Nanocomposite II	97 ± 2

**Table 4 ijms-23-05956-t004:** Toxicity of samples expressed as % tail DNA content.

Sample	% Tail DNA (Average ± sd)
Non-exposed cells 0 h	23.43 ± 8.76
Non-exposed cells stored for 24 h	25.81 ± 14.31
Control composite	16.26 ± 12.14
Nanocomposite I	6.48 ± 8.63
Nanocomposite II	9.39 ± 10.23

## Data Availability

The data presented in this study are available on request from the corresponding author.

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
