# Peer review of "The Functional and Application Possibilities of Starch/Chitosan Polymer Composites Modified by Graphene Oxide"

_ijms, 2022, doi:10.3390/ijms23115956_

Round 1

Reviewer 1 Report

The work presents functional properties of new bionanocomposites, enriched with 387 GO nanoparticles and based on a binary starch/chitosan matrix, obtained by green 388 synthesis method. The results are useful. However, several limitations need to be overcome.
1.  They should improve introduction and at the last paragraph, superiority of the paper should be defined.
2.  Comment 2: Results and discussion section was short and weak, and the data must be discussion in more detail.
3.  Numbers of references were not sufficient. 50% of presence must be provided.
4.   In Fig. 1-3, more reaction time should be needed.
5.   The image of fresh meat with the extension of time should be added.

Author Response

Authors:

Dear Reviewer,

Thank you very much for your constructive comments. We have revised and corrected our manuscript according to your suggestions. All the modifications are highlighted in blue. Below we present the itemized list of the introduced changes.

Reviewer 1:

The work presents functional properties of new bionanocomposites, enriched with GO nanoparticles and based on a binary starch/chitosan matrix, obtained by green synthesis method. The results are useful. However, several limitations need to be overcome.
1.  They should improve introduction and at the last paragraph, superiority of the paper should be defined.

Thank you for this comment. The introduction section has been revised in order to underline and explain the novelty of this work more clearly. This study describes functional properties of new bionanocomposites consisting of starch/chitosan/graphene oxide (GO) obtained by green preparation (i.e. without using toxic chemicals, aggressive or corrosive substances), that possess improved biological activities: good tolerability by human cells with concomitant antimicrobial activity. The obtained results clearly show that the nanocomposites obtained in this study may be included in active packaging. The novelty of this work was also to exploit the optical properties of graphene and apply it as an element of smart packaging.

  1.  Comment 2: Results and discussion section was short and weak, and the data must be discussion in more detail.

We corrected the text according to Reviewer suggestion.

  1.  Numbers of references were not sufficient. 50% of presence must be provided.

We added new references according to Reviewer suggestion.

  1.   In Fig. 1-3, more reaction time should be needed.

Additional analysis was performed as suggested by the Reviewer. The graphs have been changed, standard deviations for each measurement have been included, the table have been also improved.

  1.   The image of fresh meat with the extension of time should be added.

Below are pictures of meat stored for 4 days under four different foils. Since the photos were taken with a phone, their quality is not high. Therefore, we decided not to include them in the paper. For the purpose of the review, we show them here. (Please see attached file).

In addition, we have included pictures of meat stored under different films in the graphic abstract.

We appreciate for your time and effort, hoping that the correction will now find your approval. Finally, thank you very much for all your suggestions.

Best regards,

Authors

Reviewer 2 Report

The authors have submitted a state-of-the-art review " The  functional  and  application  possibilities of  starch/chitosan polymer composites modified by graphene oxide" which deals with the green synthesis of starch/chitosan polymer composites modified by graphene oxide and its impact on microflora activity. The manuscript is well structured and reads well overall, although it will need a spelling check and there are several big issues to be addressed. This study may be suitable for publication after a serious major revision.

Comments:

1- First of all, I would like to recommend authors to design a “graphical abstract” for this study to better show the whole story in a simple and informative manner.

2- The introduction is short and insufficient. It can’t convey the purpose of this study and it is needed to be carefully developed. The benefits of blending different biopolymers and their impacts o the final product should be mentioned. In the second paragraph of the introduction, a lot of information has been given especially related to chitosan however they should be supported with references.

Green Chemistry, 24(1), pp.62-101 , Carbohydrate Research, 489, p.107930.  

Moreover, In the last paragraph (novelty statement), the reason for the selection of GO for composition should be elucidated. 

3- There are several misspellings and errors in the style which should be revised. Please keep consistency in your writing style in the whole manuscript (including figures and tables). Moreover, the authors abbreviated Graphen Oxide with GO in the Abstract and the Introduction part, but later again in the rest of the manuscript, we can see the full name in line 266. Please keep consistency in using abbreviations for the rest of the manuscript and just use the abbreviation.

 4- What is the impact of composition on the color of the final product? Please provide some optical images of materials before and after composition. The bactericidal activity of materials is highly dependent on their topological and morphological structure. Please provide SEM images of materials (alone) and after GO composition as well.

 5- Bacterial adhesion is another important factor in material with antibacterial activity. What is the surface charge of your material? And to what extent GO composition has changed it? You can better explain it using surface functional groups of each material (FTIR) or any other feasible analysis.

6- I can’t see any error bar in any of the figures. To increase the reliability of your data please repeat your experiments with at least 3 replication and resketch figures with an error bar.

7- The caption for Table 2 is weird! “Table 2. This is a table. Tables should be placed in the main text near to the first time they are cited.”!!! Please revise the whole manuscript, and delete unnecessary sentences!. Please double-check figure captions as well.

8- One of the important parameters in packing material is their oxygen permeability. Please provide data related to the oxygen permeability of composite against pristine material as well.

Author Response

Authors:

Dear Reviewer,

Thank you very much for your timely and helpful suggestions. We have revised and corrected our manuscript according to your suggestions. Below we present the itemized list of the introduced changes.

Reviewer 2:

The authors have submitted a state-of-the-art review " The functional  and  application  possibilities of  starch/chitosan polymer composites modified by graphene oxide" which deals with the green synthesis of starch/chitosan polymer composites modified by graphene oxide and its impact on microflora activity. The manuscript is well structured and reads well overall, although it will need a spelling check and there are several big issues to be addressed. This study may be suitable for publication after a serious major revision.

 Comments:

 1- First of all, I would like to recommend authors to design a “graphical abstract” for this study to better show the whole story in a simple and informative manner.

 Thank you for this remark, we prepared a “graphical abstract” according to suggestion.

2- The introduction is short and insufficient. It can’t convey the purpose of this study and it is needed to be carefully developed. The benefits of blending different biopolymers and their impacts o the final product should be mentioned. In the second paragraph of the introduction, a lot of information has been given especially related to chitosan however they should be supported with references.

Green Chemistry, 24(1), pp.62-101 , Carbohydrate Research, 489, p.107930.  

The introduction section was improved according to Reviewer suggestions.

Moreover, In the last paragraph (novelty statement), the reason for the selection of GO for composition should be elucidated. 

It was added according to Reviewer suggestions.

3- There are several misspellings and errors in the style which should be revised. Please keep consistency in your writing style in the whole manuscript (including figures and tables). Moreover, the authors abbreviated Graphen Oxide with GO in the Abstract and the Introduction part, but later again in the rest of the manuscript, we can see the full name in line 266. Please keep consistency in using abbreviations for the rest of the manuscript and just use the abbreviation.

Thank you for this remark, we corrected the text according to Reviewer suggestions.

 4- What is the impact of composition on the color of the final product? Please provide some optical images of materials before and after composition. The bactericidal activity of materials is highly dependent on their topological and morphological structure. Please provide SEM images of materials (alone) and after GO composition as well.

Thank you for this comment. We carried out the SEM images and published in our previous work:

Krystyjan, M., Khachatryan, G., Grabacka, M., Krzan, M., Witczak, M., Grzyb, J., & Woszczak, L. (2021). Physicochemical, bacteriostatic, and biological properties of starch/chitosan polymer composites modified by graphene oxide, designed as new bionanomaterials. Polymers, 13(14), 2327.

 5- Bacterial adhesion is another important factor in material with antibacterial activity. What is the surface charge of your material? And to what extent GO composition has changed it? You can better explain it using surface functional groups of each material (FTIR) or any other feasible analysis.

The FTIR analyses were carried out and published in our previous work:

Krystyjan, M., Khachatryan, G., Grabacka, M., Krzan, M., Witczak, M., Grzyb, J., & Woszczak, L. (2021). Physicochemical, bacteriostatic, and biological properties of starch/chitosan polymer composites modified by graphene oxide, designed as new bionanomaterials. Polymers, 13(14), 2327.

6- I can’t see any error bar in any of the figures. To increase the reliability of your data please repeat your experiments with at least 3 replication and resketch figures with an error bar.

Thank you for this remark, we did our best to correct these issues.

7- The caption for Table 2 is weird! “Table 2. This is a table. Tables should be placed in the main text near to the first time they are cited.”!!! Please revise the whole manuscript, and delete unnecessary sentences!. Please double-check figure captions as well.

It was corrected according to Reviewer suggestions.

8- One of the important parameters in packing material is their oxygen permeability. Please provide data related to the oxygen permeability of composite against pristine material as well.

Thank you for your attention, unfortunately for time reasons we were not able to perform this analysis, but in the future, we will try to do so. This is because we plan to continue our scientific research in this direction, with the expansion of the application scope of composites.

We appreciate for your time and effort, hoping that the correction will now find your approval. Finally, thank you very much for all your suggestions.

Best regards,

Authors

Round 2

Reviewer 1 Report

OK, the manuscript can be accepted now.

Reviewer 2 Report

The manuscript is well amended and ready to be published. I have no further comments.